# Plasticity in Compensatory Growth to Artificial Defoliation and Light Availability in Four Neotropical Understory and Forest Edge Herb Species

**DOI:** 10.3390/biology11101532

**Published:** 2022-10-19

**Authors:** Jennifer W. C. Sun, M. Rasoul Sharifi, Philip W. Rundel

**Affiliations:** Department of Ecology and Evolutionary Biology, University of California, Los Angeles (UCLA), Los Angeles, CA 90095, USA

**Keywords:** defoliation, compensatory growth, understory, forest edge, broad-leave monocot herb

## Abstract

**Simple Summary:**

For monocot herbs growing in the understory of wet tropical forests face an extreme hazard in falling branches and debris from the canopy. We compared the response of two species of understory herbs to two other species of herbs growing at the forest edge or in large gaps. We made the prediction that the forest edge herb species would be better able to compensate for damage because of compensatory growth made possible in the higher light environment than that experienced by the shade-tolerant understory herbs. Our experimental studies showed that both groups of species were tolerant of defoliation under high and intermediate light conditions, but under low light growth conditions the forest edge species showed higher mortality. This finding suggests that a variety of functional growth traits may be structuring post-damage response in understory and forest edge herbs.

**Abstract:**

Defoliation from falling branches is a major factor in the survival of understory herbs in tropical rainforests. Experimental studies of defoliation under three levels of light environment compared responses to partial and total defoliation in four species of tropical rainforest understory herbs. We predicted that elevated levels of light availability would help compensate for damage to through compensatory growth in both understory and forest edge species and that forest edge species would more effectively compensate under high light conditions than shade-tolerant species from the forest understory All species showed a high tolerance to defoliation under high and intermediate light conditions. Under low-light conditions survival differed dramatically with minimal mortality in forest-edge species compared to high mortality in completely defoliated understory species. Defoliation, and light × defoliation interactions, impacted multiple growth traits in understory species. In contrast, forest-edge species showed no effect of defoliation except on total biomass, and only one light × defoliation interaction was observed. Our results indicate that differences in biomass allocation, leaf ecophysiology, and other growth parameters between forest understory and edge species may be structuring post-damage response in understory and forest edge herbs.

## 1. Introduction

Plants in the understory of tropical forests are often subjected to natural defoliation caused by falling canopy debris [1,2,3,4,5], herbivory [6,7,8], and/or pathogens [9]. Although the reduction of photosynthetically active tissue and subsequent loss of whole-plant carbon gain may reduce plant performance and fitness [10,11,12], numerous studies have demonstrated that plants are often able to mitigate the effects of defoliation through compensatory growth that plants are often able to mitigate the effects of defoliation through compensatory growth [13,14,15,16]. Thus, the effect of defoliation on plant performance can be considered as a continuum from negative to positive [17,18].

Tolerance to defoliation depends on the environmental conditions under which plants grow [16], particularly the resources available to stimulate recovery. For plants in the understory layer of tropical forests, the capacity for compensatory growth may be profoundly influenced by the wide gradient of light levels, ranging dense shade well under 1% full sun under a closed canopy to over 70% full sun in large tree-fall gaps [19,20]. Yet, areas of high light availability are rare and unpredictable in time and space [21,22]. and understory plants may persist in relatively low light conditions for a large portion of their life. The negative impacts of defoliation might be expected to decrease with increasing light availability. However, studies have also shown that many understory plants show signs of damage from photoinhibition at high light intensities [23,24]. As a result, the interactive effects of light and defoliation on understory plant performance remain unclear.

While defoliation directly reduces a plant’s light acquisition organs, it can also improve conditions such that steady rates of growth are maintained or even increased. In fact, improved availability of water and nutrients to remaining leaves of partially defoliated plants are associated with increased leaf photosynthetic capacity. At the whole-plant level, defoliation can also enhance light penetration if self-shading is reduced [11,25,26,27,28], which may be especially important for plants living in low light conditions [29].

Although the consequences of leaf damage have been well documented for herbaceous plants in temperate regions], few studies have examined their tropical counterparts [29,30,31,32] and even fewer have addressed broad-leaved monocots (but see [33,34]) Broad-leaved monocots, the dominant group of tropical understory herbs, present an ideal system to evaluate the effect of light availability in shaping post-damage recovery. They are distributed across a variety of habitats ranging from dense forest understories to full-sun pastures [35,36] and contain a wide array of ecological traits including variations in leaf physiology, morphology, architecture, life history, and vegetative reproduction [37,38,39].

Unlike woody trees, understory herbs never grow to a size where they can escape a high risk of damage and may therefore employ a different strategy for recovery. An important mechanism for compensatory growth may be their strong vegetative reproduction capability, which can support rapid growth, spread the mortality risk among ramets, and facilitate the sharing of nutrients and water to surviving tissue [38].

Furthermore, many herb species allocate a substantial portion of their biomass to rhizomes [34,39] which can store carbohydrates for use in tissue replacement.

We selected four rhizomatous herb species common to lowland rainforests of Costa Rica selected to be phylogenetically independent examples of shade vs. forest edge habitats. *Goeppertia micans* (Figure 1a) and *Heliconia irrasa* (Figure 1b) are shade-tolerant herbs found in the understories of primary and secondary forests. *Goeppertia micans* is a low-growing herb, reaching 10 cm in height, and is found in deeply shaded conditions. In contrast, *H. irrasa* commonly reaches 1–1.5 m in height and is found in shaded understory habitats as well as small light gaps where they persist as clonal stands (J. Sun, personal observation). The remaining two species, *Goeppertia marantifolia* (Figure 1c) and *Costus malortieanus* (Figure 1d), are moderately-sized herbs (reaching 2.3 m and 1 m, respectively) and require higher light levels for growth and reproduction. Both species occur along forest edges, but *C. malortieanus* is a frequent colonizer of canopy gaps and shrubby secondary growth.

All four species reproduce both sexually and well as clonally (Figure 2). Leaf damage caused by falling canopy debris and parasitic associations with herbivores [40] is a common and constant factor for all four species.

Our objective in this project was to utilize experimental growth studies to test two hypotheses. The first is the prediction that elevated levels of light availability will help compensate for damage to stimulate rapid compensatory growth in both understory and forest edge species. The second is that forest edge species, which characteristically are exposed to high levels of irradiance during some part of the day, will more rapidly compensate through compensatory growth for biomass lost by defoliation under high light conditions than shade-tolerant species from the forest understory.

## 2. Materials and Methods

### 2.1. Study Site and Species

This study was conducted at the La Selva Biological Station in northeastern Costa Rica from November 2002 to July 2003. The natural vegetation is classified as Tropical Wet Forest (sensu Holdridge [41]) with an average annual temperature of 26.5 °C (range 20–31 °C). The research station has a mean annual rainfall of 4244 mm (1958–2004), with mean monthly rainfall above 300 mm from May through December. There are peaks of precipitation above 400 mm mo^−1^ in June-August and November-December, and a drier period from January through April. Even in the driest period of February and March, however, rainfall averages above 150 mm each month. Mean monthly maximum and minimum temperatures show very little seasonal change, with mean highs of 30–31 °C each month and mean monthly lows ranging only from 20–22 °C.

We selected two common herb species representing shaded understory habitats to compare with two species of forest edge habitats in response to differing levels of defoliation and light environment. Our experimental growth studies focused on survival of parent ramets, not through sexual reproduction Our study included four rhizomatous herb species common to lowland rainforests of Costa Rica. The understory species were *Goeppertia micans* (L. Mathieu) Borch. & S. Suarez (Marantaceae) and *Heliconia irrasa* R.R. Smith subsp. *undulata* Daniels and Stiles (Heliconiacae). The two forest edge species were *Costus malortieanus* H. Wendl. (Costaceae.) and *Goeppertia marantifolia* (Standley) Borch. & S. Suarez (Marantaceae). These are common perennial broad-leaved herbs found throughout lowland rainforests in Costa Rica. Forest understory species, such as *G. micans* and *H. irrasa*, are generally more shade-tolerant, slow growing, and less plastic in photosynthetic light response than forest edge species [38,42]. The demography and reproductive biology of *Goeppertia marantifolia* has been studied extensively [43,44,45,46].

### 2.2. Experimental Methods

In November 2002, we collected 90 individuals of each study species from natural populations across a broad area of the La Selva primary forest understory. Plants were potted into 3.8 L pots filled with homogenized local soil and left in their original habitat for two months to diminish uprooting and potting stress. We randomly assigned each plant to one of three light treatments (2, 10 and 50% full irradiance) in a partitioned shadehouse created with neutral-density shade cloths. Light levels roughly correspond to natural conditions experienced by herbs in the understory (2% full sun), late succession canopy gaps or forest edges (10%), and early succession open areas (50%).

After the plants acclimated to the light treatment for 30 days, we randomly assigned each plant to one of three leaf defoliation treatments: 0% (control), 50% (partial defoliation), or 100% (complete defoliation). We selected a broad range of defoliation values based on our previous observations at La Selva, where damage and herbivory can remove up to 100% of laminar leaf area in understory herbs (J. Sun, personal observations). Within each replication of light availability, we placed ten individuals for each defoliation level per species, for a total of 360 individuals in the experiment. To control for nutrient availability, each individual was given liquid fertilizer (Bayfolan Forte) at the beginning of each month.

### 2.3. Physiological and Morphological Measurements

In July 2003, after eight months of growth, we recorded maximum photosynthetic rate on two to three recently expanded and matured leaves from multiple individuals for each species in each light treatment using a LI-COR 6400 portable photosynthesis system (LI-COR Inc., Lincoln, Nebraska) equipped with a CO_2_ control module and a red-blue light emitting diode light source (Model 6400-02B). The ambient temperature inside the leaf chamber was kept at 26 °C, close to the maximum ambient daytime temperature when the measurements were made. The leaf-to-air vapor pressure deficit (VPD) was maintained at 0.5 kPa. The CO_2_ concentration inside the leaf chamber was kept constant at 400 mmol mol^−1^ during the light response curves. Gas exchange characteristics and associated leaf properties were determined for leaves that had developed in each light regime under conditions of partial defoliation.

Plants were harvested and separated into leaves, petioles, and roots. We used a LI-3100 leaf area meter (LI-COR Inc., Lincoln, NE, USA) to measure the photosynthetic surface area of each leaf after carefully trimming blade portions from the midvein. Plant parts were dried to a constant mass at 70 °C for 72 h and weighed to the nearest 0.001 g using a portable balance (Ohaus Navigator, Pine Brook, NJ, USA). Using harvest measurements, we calculated the following biomass- and growth-related variables: total biomass, leaf mass ratio (LMR), stem mass ratio (SMR), root mass ratio (RMR), root to shoot ratio (RSR), specific leaf area (SLA), and number of vegetative shoots. We also calculated the leaf production rate (LPR) relative to each plant’s initial number of leaves to standardize for differences in leaf number among individuals. Overall plant survival was noted prior to harvesting.

### 2.4. Statistical Analysis

We analyzed the effects of defoliation and light availability on growth indices in all four species using a split-plot ANOVA, where light effect was the main plot and the defoliation effect the subplot. This design enabled us to not only examine the main effects of light or defoliation on plant performance, but also the light × defoliation interaction. We used post hoc Tukey HSD tests to compare treatment level means for significant factors in ANOVA. To meet the assumptions of normality, we used the function ln(z) to transform all variables and ln (z + 2) for leaf production rate. Survival was analyzed with a logit analysis, in which light, defoliation, and species were introduced as independent variables and plant survival as the response variable. All statistical analyses were performed in SPSS 17.0 [47].

## 3. Results

### 3.1. Survival

Overall percentage of survival was high for this study (96.1%). In high and intermediate light conditions, survival over 8 months was not significantly affected by the removal of 50% or even 100% of leaf area for all species (Figure 3). In low light conditions, survival was similarly high in all but completely defoliated plants. Whereas all complete defoliated plants of forest edge species survived under low light (Figure 3a), complete defoliation killed 40% of understory species, the lowest overall percentage of survival in this study (Figure 3b; χ^2^ = 38.02, *p* < 0.001).

### 3.2. General Light and Defoliation Effects

Across all four species, the effect of light environment was significant for most traits, with differences being driven by the high-light treatment (Table 1; Figure 4 and Figure 5). Herbs were generally larger, produced more leaves and vegetative shoots, and had lower specific leaf area (SLA) when grown under high light levels. The only traits that did not show significant variation with light were stem mass ratio in *Goeppertia micans*, allocational variables in *Costus malortieanus*, and leaf production rate in *Goeppertia marantifolia* (Table 1). With the exception of total biomass, the effect of defoliation was only significant for understory species subjected to 100% leaf removal. In *G. micans* notably, complete defoliation affected all variables but SLA (Table 1).

### 3.3. Biomass Allocation

We found consistent patterns in biomass allocation between both forest edge and understory species. Total biomass increased with elevated light availability and lower intensity of defoliation, though the increase was much greater in forest edge than understory species (Figure 4a–d). Among control plants, total biomass was highest for *Goeppertia marantifolia* (range 15.438 ± 2.304 g in low light to 59.098 ± 4.373 g in high light; Figure 4a) and lowest in *G. micans* (0.709 ± 0.080 in low light to 3.745 ± 0.402 g in high light; Figure 4c). Whereas partial defoliation did not result in any changes to overall biomass accumulation, completely defoliated plants had significantly lower total biomass than control plants (Figure 4a–d).

Light had a strong effect on the leaf mass ratio (LMR), root mass ratio (RMR). and root to shoot ratio (RSR) for all species but *Costus malortieanus*. While mean values of LMR in understory species was more than double that of forest edge species (Figure 4e–h), the reverse was true for mean values of RMR (Figure 4m–p). SMR was relatively similar for all species, but *Goeppertia marantifolia* invested more biomass into stems and petioles than the other three species (Figure 4i). The effect of defoliation and the light × defoliation interaction on LMR, RMR, and RSR was only significant for shade-tolerant species, where completely defoliated plants increased leaf production at high light levels, likely at the expense of below-ground structures (Table 1; Figure 4g,h).

### 3.4. Shoot Production

Shoot production and leaf structure did not vary predictably with habitat type. Specific leaf area showed a negative response to light availability and no response to defoliation in all four species (Figure 5a–d). For plants grown under high light conditions, leaf production rate (LPR) was significantly higher in all species but *Goeppertia marantifolia* (Figure 5e–h). The effect of defoliation on LPR was only detected in *G. micans*, where completely defoliated plants produced leaves at a slower rate than control or partially defoliated plants (Figure 5g). Notably, we detected a highly significant light × defoliation interaction for LPR in *Heliconia irrasa*, where plants with complete defoliation increased leaf production at high light levels (Table 1). Similar to leaf production rate, the number of vegetative shoots was greatest in high light conditions for all four species (Figure 4i–l). *Goeppertia micans* also had the greatest number of vegetative shoots across all treatments (Figure 5k), though this is not an accurate indicator of clonal ability in the other three species as pot size likely limited clonal expansion. Whereas defoliation had no effect on the number of vegetative shoots in forest edge species, it had a negative effect on vegetative shoot production in completely defoliated individuals of understory species (Table 1; Figure 5i–l).

### 3.5. Photosynthetic Responses

Assimilation rates were highest for *Goeppertia marantifolia* and *Heliconia irrasa* under low and intermediate light treatments, and highest for *Costus malortieanus* under high light treatment (Table 2). Relative increases in assimilation rates and saturating PFD intensities across light treatments were similar for leaves of *C. malortieanus, G. marantifolia*, and *H. irrasa*, with light saturation occurring at 800 µmol m^−2^ s^−1^ in leaves adapted to high light conditions. *Costus malortieanus* showed the greatest physiological plasticity to varying light conditions, with light saturation occurring from 150 µmol m^−2^ s^−1^ in low light to 800 µmol m^−2^ s^−1^ in high light. Assimilation rates for high light leaves of *C. malortieanus* were also 3.5 times higher than leaves produced in low light. In contrast, light saturation for *Goeppertia micans* only showed a slight increase in saturating PFD, ranging from 250 µmol m^−2^ s^−1^ in low light to 400 µmol m^−2^ s^−1^ in high light. The assimilation rate for leaves of *G. micans* produced under high light was only 4.56 ± 0.19 µmol m^−2^ s^−1^, which is half the rate recorded in the other three species.

## 4. Discussion

Tolerance to defoliation is typically associated with compensatory growth, a mechanism through which negative effects of leaf loss are mitigated. There are a variety of possible strategies to achieve recovery from defoliation [14]. One of these is to preferentially allocate new assimilates to leaf production, A second possible strategy is to make more efficient use of photosynthetic leaf surface by increasing specific leaf area. Finally, plants may increase their rates of net photosynthetic assimilation thereby increasing growth rates [42,48,49].

Our results indicate that tropical broad-leaved herbs exhibit a high tolerance to leaf loss, and opportunities for post-damage compensatory growth are greater at increasing light levels. Plants grown in progressively higher light conditions accumulated more biomass and had a greater likelihood of survival than plants grown in lower light levels, as has been also shown in previous studies [50,51,52]. Although the dramatic increase in maximum assimilation rates by a factor of 3.6 for *Costus malortieanus* is higher than the value of 2.5 reported by Bazzaz and Carlson [53] and 2.6 reported by Sims and Pearcy [54] for other tropical herb species, these rates are similar to the increased rates of 2.3 in *H. irrasa* and 2.2. in *Goeppertia marantifolia*. For the understory species, *Heliconia irrasa* exhibited a much higher level of plasticity toward light manipulation than *C. micans*, with maximum assimilation rates ranging from 4.35 ± 0.30 to 10.20 ± 0.58 µmol m^−2^ s^−1^ and light saturation occurring at 800 µmol m^−2^ s^−1^ in leaves produced under high light conditions. This may be explained by the fact that *H. irrasa* is substantially larger than *Goeppertia micans*, and larger plant size is often associated with the ability to allocate more carbon to the production of new leaves with improved photosynthetic capacities [55,56]. Furthermore, the high costs of maintaining phenotypic plasticity may not pay off for the low-growing *G. micans* as they are easily overtopped by other plants and persist in perpetually low light conditions.

Indeed, maintaining or increasing rates of growth after partial defoliation has been widely exhibited in tropical understory palms [30,37,57,58,59,60], seedlings [5,61], and a few broad-leaved herb species [33,34]. Greenhouse studies on the understory palm *Chamaedorea elegans* Mart. found that improved light penetration following defoliation was strongly associated with increased photosynthetic capacity in leaves and whole-plant carbon gain [11]. Shifts in biomass allocation to leaf production in the understory palm *Geonoma deversa* (Poit.) Kunth. at the cost of stem structures was shown to be critical in the rapid recovery of leaf area [62]. Thus, maximizing light harvesting capacity may be more effective than strategies favoring structural support, especially during times of increased light availability as with canopy gap formation.

Other studies have also shown that defoliation can improve growing conditions by increasing the water and nutrient supply available for remaining leaves [63,64]. The high survival rates of partially defoliated plants in our study also suggest that leaf loss may have a much greater effect on growth rates than survival in understory herbs, which until now, has only been demonstrated for long-lived plants with slow growth and high survival rates [65,66,67].

While partial leaf area removal was tolerated extremely well across all species, the effect of complete defoliation varied by habitat. Overall, plants subjected to complete leaf removal in our study accumulated significantly less biomass than partially defoliated or control plants. In the understory species *Goeppertia micans* however, severe damage also resulted in increased mortality at low light and lower rates of leaf and shoot production than control or partially defoliated plants. These results indicate that the sudden removal of all photosynthetic organs may have exceeded the damage threshold for *G. micans* under low light conditions. We also detected a significant light × defoliation effect for LMR, RMR, RSR, and LPR in both forest understory species. The understory plants that survived severe damage were able to increase leaf production at elevated light levels, which may be an important compensatory mechanism in ameliorating the negative effects of severe leaf loss [31,68].

Biomass allocation to belowground tissues may be an important component of resilience to defoliation. Previous studies have demonstrated that investing in root systems not only reduces the risk of above-ground damage, but also maintains a carbohydrate and water reserve that can be mobilized for the reconstruction of new leaves [69,70] or shared among remaining leaves [64,71,72,73]. Indeed, plants with high pre-defoliation levels of stored carbohydrates have been shown to grow more rapidly and have higher probabilities of survival post-damage than those with lower levels [74,75]. In our work we did not directly measure the interaction of defoliation with belowground resources and storage capacity in rhizomes, which we believe is likely a strong determinant of compensatory abilities for the species in this study [75,76].

Our results indicate that differences in above- and belowground biomass allocation, leaf physiology and other growth parameters between forest understory and edge species may be structuring post-damage response in understory herbs. Although all species showed a high tolerance to defoliation damage, forest edge species were able to maintain steady rates of growth, even after total removal of leaf area, with an increased photosynthetic capacity for leaves produced in high light conditions. A study of 14 broad-leaved herb species at La Selva found that forest edge species can afford to invest less carbon in structural reinforcement than species in the shaded understory because rapid and clonal growth is a potentially more effective mechanism in risk recovery [38]. Indeed, studies on tropical understory plants have found less foliar damage in light-demanding, fast-growing pioneer species than shade-tolerant understory species [2,77,78,79,80,81,82].

Experimental studies with *Goeppertia marantifolia* by other researchers provide supporting data on the complexity of interpreting growth response to light levels. Growth response to gradients of light level have been carried out comparing growth and survival clonal offshoots and seedlings to gap, gap edge, and forest understory microsites [43,44,45,46]. Seedlings were sensitive to light availability and survived best in gap centers, while vegetative offspring had their highest survival in the shaded understory. These results suggested that being attached to the parent provided buffering for vegetative offspring with respect to survival, but that higher growth rates are associated with independence and increased risk of mortality. The magnitude of the cost of reproduction and trade-offs between reproductive modes was small and there was no evidence of lower costs of reproduction for larger plants or for plants in higher light environments. Light availability was positively associated with clonal dispersal distance indicating the significance of light resources for clonal dynamics., the influence of resource availability on spatial population dynamics [43].

Experimental studies with variable light environments carried out with two understory relatives to species in our study, *Calathea crotalifera* and *Heliconia tortuosa*, are relevant to our results. These findings suggest that generalist understory species may experience increased fitness under variable light conditions by maintaining high growth at the expense of survival. Differences among species in their abilities to thrive under variable light conditions and thus occupy distinct niches may contribute to the maintenance of species diversity Through their effects on growth photosynthetic rates may strongly influence the population dynamics of plants in variable light environments, but the magnitude of this effect varies between species [83,84,85].

## 5. Conclusions

Consistent with what we predicted, elevated levels of light availability help compensate for damage through compensatory growth in both understory and forest edge species. We had mixed results with our prediction that forest edge species would more effectively compensate under high light conditions than shade-tolerant species from the forest understory. The greater capacity for recovery in forest edge species may be due to several factors. First, forest edge species allocated a greater portion of their biomass to belowground structures than shaded understory species. Second, forest edge species exhibit more clonal growth in the field which can facilitate rapid growth while spreading the mortality risk among ramets [30,38]. Clonal plants can also share resources between ramets such that steady rates of growth are maintained after losing photosynthetic organs [30]. Lastly, forest-edge species exhibited high levels of plasticity in leaf physiology traits toward light manipulation. This suggests that they are able to efficiently use higher light conditions, which may be a potentially important mechanism for compensatory growth.

In contrast to the effect of light, we did not detect any differences in growth- or biomass-related variables between defoliated and control plants for forest edge species and between partially defoliated and control plants for forest understory species. Although one might conclude that damaged plants did not compensate for the losses caused by defoliation, even the maintenance of values similar to control plants requires compensatory response [31,32].

Genetic plasticity may be important in understanding plant responses to defoliation [49,59]. Complicating this issue is the fact that most trait-based ecological studies analyze mean values of species traits, obscuring relationships caused by variation in biotic and abiotic environmental factors [39]. Traits we observed in *Costus malortieanus* may be particularly effective in responding to defoliation. Damaged plants grown under high light had greater total biomass, more vegetative shoots, higher leaf production rates, and thicker leaves with greater photosynthetic rates than control plants at low light.

Leaf physiological traits also exhibited remarkable plasticity, with maximum assimilation rates more than three times higher in high light than low light treatment. Our light response curves suggest that different understory monocots exhibit different levels of plasticity toward PAR manipulation as the forest edge *Goeppertia marantifolia* and *Costus malortieanus*, as well as the understory *Heliconia irrasa* are more plastic in response than *G. micans*. This latter species is less flexible in responding to increasing PAR than other co-occurring species suggesting that it lacks the potential photosynthetic capacity to take advantage of increased light (at high or intermediate light). It exhibited low light saturation and maximum photosynthetic rate under high light treatment is only slightly higher than that measured in shaded understory conditions.

Unlike the other species in this study, *C. malortieanus* has a growth form similar to woody pioneer seedlings such as *Cecropia*, with leaves spirally arranged atop a single stem to maximize leaf area index. Leaves of *C. malortieanus* are also broadly obovate, born on short petioles, and significantly thicker than leaves in the other three species. Thicker leaves may be indicative of extra layers of palisade or longer palisade cells which enhances the photosynthetic capacity per unit leaf area by supporting more chloroplasts and photosynthetic enzymes [81]. In contrast, *G. marantifolia* allocated a greater ratio of carbon to petioles and stems than leaves or roots and showed no difference in leaf production rate for either light or defoliation effect. Although both species occur in higher light environments, it appears that *C. malortieanus* has traits adapted more for light harvesting and *G. marantifolia* more for height expansion and leaf support.

## Figures and Tables

**Figure 1 biology-11-01532-f001:**
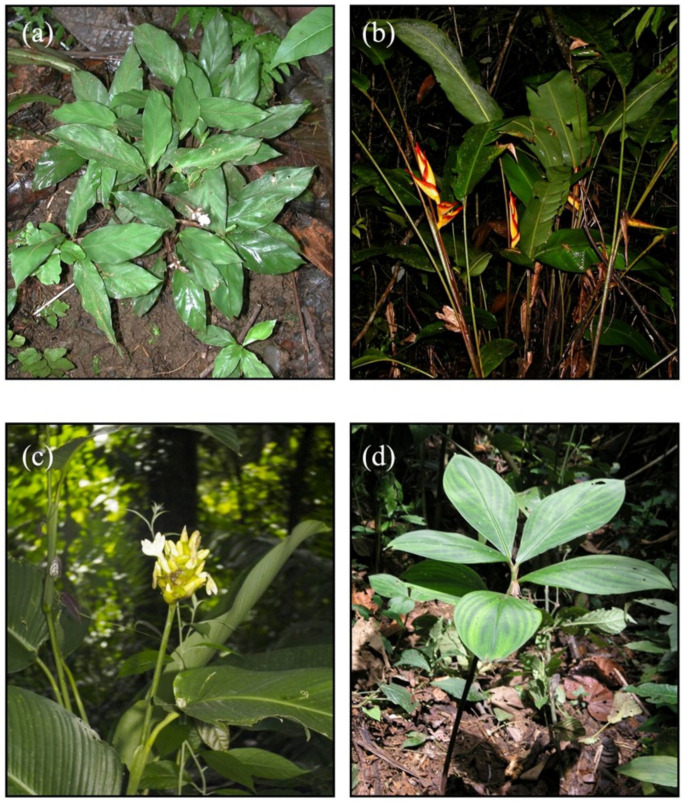
Understory herb species common to the understories of neotropical rain forests: (**a**) *Goeppertia micans* and (**b**) *Heliconia irrasa* occur in forest understory habitats, while (**c**) *Goeppertia marantifolia* and (**d**) *Costus malortieanus* occur in forest edge habitats.

**Figure 2 biology-11-01532-f002:**
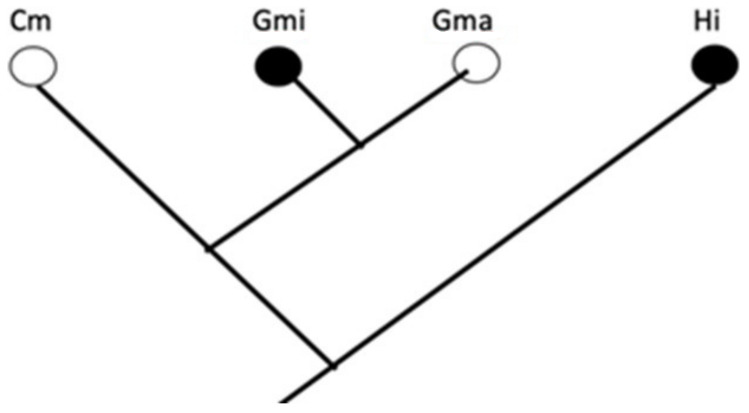
Phylogenetic relationships of the four study species.

**Figure 3 biology-11-01532-f003:**
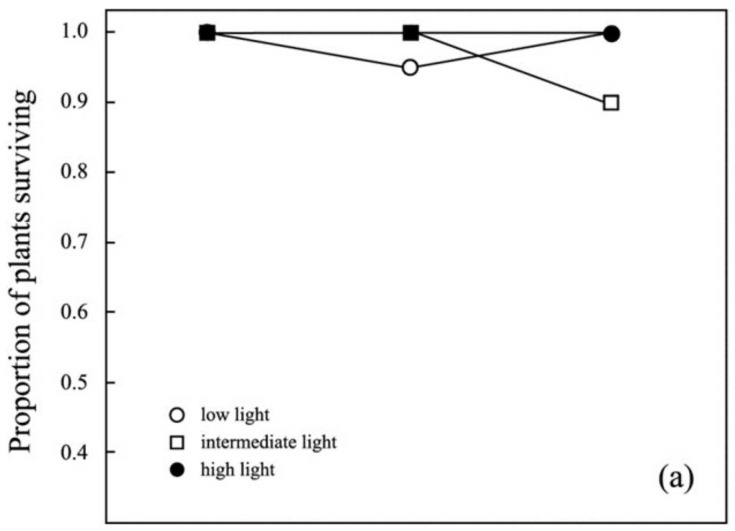
Effects of artificial defoliation treatments on the survival of understory herbs in different light levels for forest edge species *Costus malortieanus* and *Goeppertia marantifolia* (**a**) and forest understory species *Goeppertia micans* and *Heliconia irrasa* (**b**). See text for details of experimental treatments.

**Figure 4 biology-11-01532-f004:**
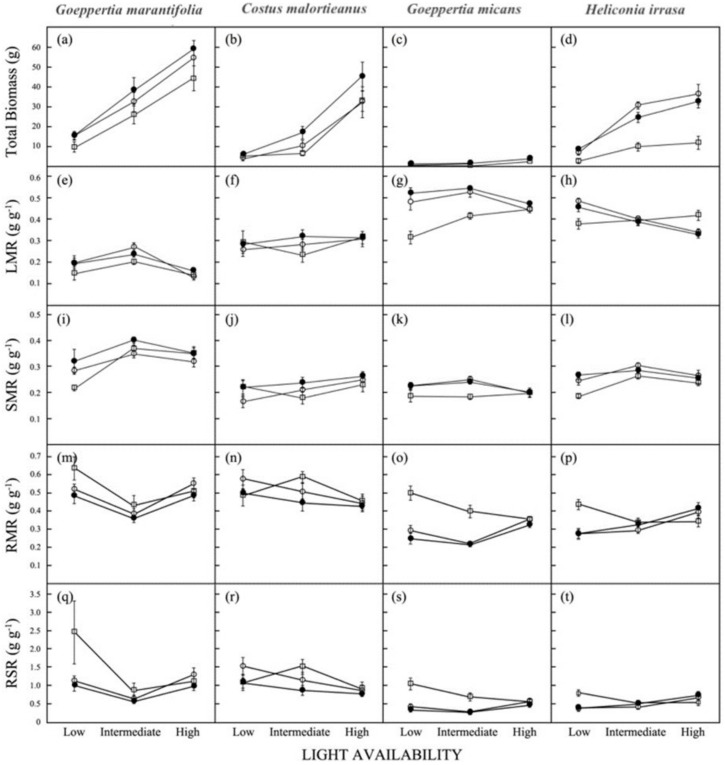
Effects of light availability (low, intermediate, and high) and defoliation (closed circle = control, open circle = 50%, and open square = 100% leaf removal) on biomass allocation parameters for four understory herb species: total biomass (**a**–**d**), leaf mass ratio (LMR, (**e**–**h**)), stem mass ratio (SMR, (**i**–**l**)), root mass ratio (RMR, (**m**–**p**)), and root-to-shoot ratio (RSR, (**q**–**t**)). Values are mean ± 1 SE.

**Figure 5 biology-11-01532-f005:**
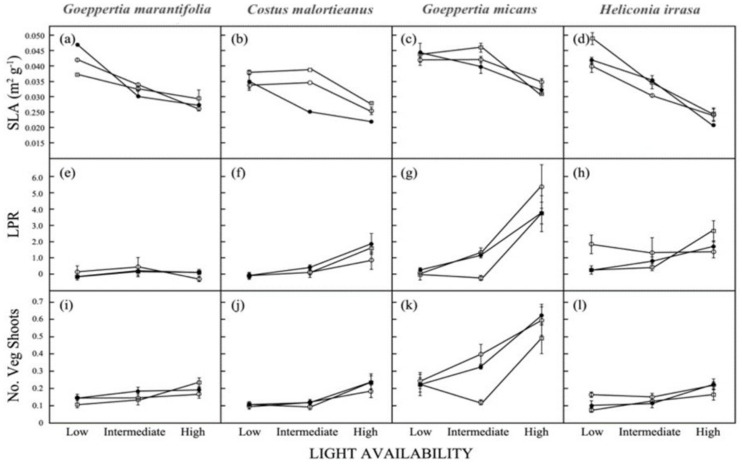
SLA, (**a**–**d**), leaf production rate (LPR, (**e**–**h**)), and number of vegetative shoots (**i**–**l**) for four understory herb species. Values shown are mean ± 1 SE.

**Table 1 biology-11-01532-t001:** Results from the split-plot two-way ANOVA using defoliation (control, 50% and 100% leaf removal) and light availability (low, intermediate and high) as fixed factors affecting metrics of growth for four broad-leaved understory species. Data were collected at the end f the experiment for total biomass, leaf mass ratio (LMR), stem mass ratio (SMR), root mass ratio (RMR), root to shoot ratio (RSR), specific leaf area (SLA), leaf production rate (LPR), and number of vegetative shoots. Analyses were conducted separately for each species. Asterisks show levels of significance in *p* values; * *p* < 0.05, ** *p* < 0.01, *** *p* < 0.001.

Dependent Variable	Light	Defoliation	Light × Defoliation
*Goeppertia marantifolia*			
Total Biomass	66.44 ***	6.15 *	0.30
LMR	11.88 ***	2.40	0.74
SMR	7.92 **	2.32	1.68
RMR	14.49 ***	2.44	0.78
RSR	11.83 ***	3.03	1.20
SLA	41.72 ***	1.54	0.28
LPR	1.64	0.12	1.08
No. Vegetative Shoots	8.53 ***	0.45	2.68 *
*Costus malortieanus*			
LMR	1.56	1.06	0.99
SMR	2.98	1.31	1.05
RMR	2.51	1.06	1.26
RSR	3.05	1.43	1.35
SLA	24.38 ***	8.41	2.19
LPR	20.38 ***	1.71	0.90
No. Vegetative Shoots	20.754 ***	0.64	1.05
*Goeppertia micans*			
Total Biomass	100.98 ***	34.40 ***	2.28
LMR	8.36 ***	36.39 ***	9.42 ***
SMR	2.07	7.95 ***	1.51
RMR	10.11 ***	33.73 ***	6.78 ***
RSR	8.58 ***	34.01 ***	6.98 ***
SLA	33.92 ***	0.24	0.15
LPR	76.58 ***	7.52 **	2.41
No. Vegetative Shoots	17.15 ***	12.19 ***	2.33
*Heliconia irrasa*			
Total Biomass	70.66 ***	46.47 ***	0.572
LMR	11.66 ***	0.62	6.93 ***
SMR	11.65 ***	8.92 **	2.24
RMR	4.81 *	3.16 *	5.30 ***
RSR	4.89 **	3.11	5.53 ***
SLA	58.10 ***	1.92	1.38
LPR	7.96 ***	1.24	3.02 *
No. Vegetative Shoots	7.69 **	8.37 ***	1.06

**Table 2 biology-11-01532-t002:** Photosynthetic parameters of the light response curve of photosynthesis for four understory herbs grown in varying light conditions (low, intermediate, high). Measurements were made at leaf temperature between 25 to 26 °C and leaf to air vapor pressure deficits of <1.0 MPa. Values are averages of 2–3 leaves from each of 2–3 different individuals ± 1 SE.

Dependent Variable	Low	Intermediate	High
*Goeppertia marantifolia*			
Amax (μmol m^−2^ s^−1^)	4.49 ± 0.38	6.91 ± 0.15	9.80 ± 0.30
Respiration rate (μmol m^−2^ s^−1^)	−0.31 ± 0.05	−0.33 ± 0.02	−0.50 ± 0.08
Light compensation point (μmol photons m^−2^ s^−1^)	2	4	7
Saturating PFD (mmol photons m^−2^ s^−1^)	300	600	800
Apparent quantum yield (mole CO_2_ mole photon^−1^)	2.83 ± 0.39	4.86 ± 0.51	0.04 ± 0.46
*Costus malortieanus*			
Amax (μmol m^−2^ s^−1^)	3.25 ± 0.08	5.20 ± 0.23	11.70 ± 0.22
Respiration rate (μmol m^−2^ s^−1^)	−0.35 ± 0.02	−0.35 ± 0.01	−0.59 ± 0.02
Light compensation point (μmol photons m^−2^ s^−1^)	2	5	7
Saturating PFD (mmol photons m^−2^ s^−1^)	150	600	800
Apparent quantum yield (mole CO_2_ mole photon^−1^)	0.07 ± 0.01	0.07 ± 0.01	0.02 ± 0.04
*Goeppertia micans*			
Amax (μmol m^−2^ s^−1^)	2.55 ± 0.19	2.90 ± 0.21	4.56 ± 0.19
Respiration rate (μmol m^-−2^ s^−1^)	−0.23 ± 0.02	−0.26 ± 0.05	−0.32 ± 0.03
Light compensation point (μmol photons m^−2^ s^−1^)	4	4	5
Saturating PFD (mmol photons m^−2^ s^−1^)	250	300	400
Apparent quantum yield (mole CO_2_ mole photon_-1_)	0.06 ± 0.01	0.05 ± 0.01	0.04 ± 0.00
*Heliconia irrasa*			
Amax (μmol m^−2^ s^−1^)	4.35 ± 0.30	7.88 ± 0.06	10.20 ± 0.58
Respiration rate (μmol m^−2^ s^−1^)	−0.18 ± 0.13	−0.23 ± 0.02	−0.34 ± 0.08
Light compensation point (μmol photons m^−2^ s^−1^)	4	5	7
Saturating PFD (mmol photons m^−2^ s^−1^)	250	500	800
Apparent quantum yield (mole CO_2_ mole photon^−1^)	0.07 ± 0.00	0.01 ± 0.01	0.06 ± 0.01

## Data Availability

All data collected are shown as mean values with sample sizes and standard errors shown. Complete data sets of individual measurements are available from the corresponding author.

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
