# Peer review of "Plasticity in Compensatory Growth to Artificial Defoliation and Light Availability in Four Neotropical Understory and Forest Edge Herb Species"

_biology, 2022, doi:10.3390/biology11101532_

Round 1

Reviewer 1 Report

The manuscript provides some novelty by combining plasticity and compensatory growth. However it is not adequate for publishing without careful revision. And these are my concerns:

1) Section 2.3 and 2.4 are completely the same. And the calculations of some parameters are not stated clearly: I am familiar with LMR, SMR, RMR, RSR and SLA, but confused with LPR. Is LPR (line 175) equal to the ratio between numbers of leaves in harvesting and that right after defoliation?

2) The results of ANOVA and multiple comparisons are not shown in the manuscript. Therefore it is hard to decide whether some statements in section 3 are correct or not. For example in fig 4q, RSR is significant in defoliation*light interaction for G. marantifolia base only on mean value and error bar, making statement line 257 inaccurate.

3) The caption of figure 5 is incomplete. Also the phrase “four understory herb species” (line 277) is confusing because there are 2 understory species and 2 forest edge species according to text in other place.

4) There is no Table 1.

5) The conclusion section is not concise. For example line 420-429 are better in discussion in my opinion.

6) From the References I can imagine the authors do lots of work searching through the publications. In consideration that half of the citations were published two decades ago, I strongly recommend a reduction for the references.

Author Response

GENERAL COMMENTS:

Better focus of introduction for clarity has been made and number of references has been sharply reduced..

Clarification of he research design has been made. The missing Table 1 has been added. This also impacts the methods to make them clear.

The error in double loading of the same text in sections 2.3 and 2.4 has been corrected

LPR is calculated as a ratio of leaves at the end of the experiment to its end with values normalized to account for different numbers of leaves present at the initiation.

Table 1, now present,  shows the results of the split-plot two-way ANOVA using defoliation and light availability as fixed factors affecting the metrics of growth for the four study species. We hope the presence of this table resolve questions about statistical statements in section 3.

The captions of all figures have been checked to clarify the inclusion of two forest understory species and two forest edge species in the experiments. This point has also been clarified in the text as needed.

As described above, Table 1 is now present. It was inadvertently omitted in a change to a editable format from an image.

Some reorganization of the Conclusions section has been made to make this section more concise and to separate  core  results from broader conclusions.

In response to suggestions from both the editor and Reviewer 1, we have pruned the list of references to remove less important older references.

Reviewer 2 Report

In this study, leaf-leaf experiments were carried out under three light environments. The study compared the responses of two tropical rainforest understory herbs to partial and total above-ground defoliation. The final results of the study showed that, Increased light would help understory and forest edge species compensate, and forest edge species compensated more effectively under high light conditions than understory shade-tolerant species. Survival differences were significant under low-light conditions, with forest margin species having the lowest mortality rates and fully deciduous understory species having high mortality rates. The article is of great research significance, the method is conventional, and it lacks certain innovation.

1.       It is recommended to reorganize the literature review logic in the introduction, cover the important and latest literature related to the topic, and add the innovations of this paper compared with the predecessors.

2.       Tables 1 and 2 in the text should be three-line tables.

3.       In the discussion section, it is recommended to cite the most recent research literature.

4.       The final conclusion part of the paper can summarize the research content of the full text.

5.       It is recommended to include the innovations of this paper in the Discussion section.

Author Response

Both reviewers and the editor strongly suggested that less important older reference should be pruned. This has been done with the major impacts on the introduction and discussion sections. This effort better clarifies the innovations of our paper compared with previous research.

Tables 1 and 2 are now reformatted as editable tables in response to the  editors.

As described above, the discussion section now cites the most recent research findings and better focuses the significance of our research. This addresses the last two suggestions of Reviewer 2: First, to have the final conclusions description better summarize the research content of the full text and second to include innovations in the discussion section.